# Topological and Geometrical Properties of *k*-Symplectic Structures

**Essabab Said \*, Fanich El Mokhtar**  **and Awane Azzouz**

Laboratory of Analysis, Modeling and Simulation (LAMS), Department of Mathematics and Computer Science, Faculty of Sciences Ben M'Sik, Hassan II University of Casablanca, P.O. Box 7955, Casablanca 20670, Morocco; elmokhtar.fanich-etu@etu.univh2c.ma (F.E.M.); awane.awane@gmail.com (A.A.)
\* Correspondence: said.essabab@emines.um6p.ma

**Abstract:** We study new geometrical and topological aspects of polarized *k*-symplectic manifolds. In addition, we study the De Rham cohomology groups of the *k*-symplectic group. In this work, we pay particular attention to the problem of the orientation of polarized *k*-symplectic manifolds in a way analogous to symplectic manifolds which are all orientable.

**Keywords:** *k*-symplectic; polarization; De Rham cohomology; orientation



## 1. Introduction

From 1975, the French academician Andre Lichnerowicz defined and studied several generalizations of symplectic manifolds: canonical manifolds, Poisson manifolds, Jacobi manifolds, and locally conforming symplectic manifolds. With regard to the structures related to the differential systems, A. Lichnerowicz pointed out the mechanics of Y. Nambu. In this perspective, the *k*-symplectic structures have been introduced to give a formalism to the mechanics of Y.Nambu by analogy to the symplectic geometry that constitutes the natural formalism of classical mechanics. With regard to the geometry of polarization, this notion plays an important role in the theory of geometric quantization of Kostant–Souriau; see for example [1]. It was in 1965 that J.M. Souriau gave a rigorous mathematical foundation to the process of quantification of a classical mechanical system: he used a polarization of a symplectic manifold to construct a Hilbert space and to associate, with each observable classical of a mechanical system, a self-adjoint operator on this space [2].

The *k*-symplectic structures were introduced for the first time by A. Awane [3] (1984) whose goal was to set up a formalism of the mechanics of Y. Nambu [4] in analogy with Hamiltonian mechanics.

A *k*-symplectic manifold is a triplet $(M, \theta, \mathfrak{F})$ in which $M$ is a differentiable manifold of dimension $n(k+1)$ and $\theta = \theta^p \otimes v_p \in \mathscr{A}^2(M) \otimes \mathbb{R}^k$ is a closed $\mathbb{R}^k$-valued differential 2-form and $\mathfrak{F}$ is an *n*-codimensional foliation vanishing $\theta$. The fundamental example is given by the Whitney sum $M = T^*V \oplus \ldots \oplus T^*V$ equipped with the vectorial differential 2-form $\theta = \theta^p \otimes v_p \in \mathscr{A}^2(M) \otimes \mathbb{R}^k$ subordinate to the Liouville 1-form on $M$, and $\mathfrak{F}$ is defined by the fibration:

$$\pi : T^*V \oplus \ldots \oplus T^*V \longrightarrow V.$$

The theorem of Darboux type with respect to *k*-symplectic structures [5] shows that in the neighborhood of each point of $M$, we can find a system of local coordinates $(x^{pi}, y^i)_{1 \leq p \leq k, 1 \leq i \leq n}$ such that

$$\theta = \sum_{p=1}^{k} \left( \sum_{i=1}^{n} dx^{pi} \wedge dy^i \right) \otimes v_p \tag{1}$$

and $\mathfrak{F}$ is defined locally by the equations

$$dy^1 = 0, \ldots, dy^n = 0.$$

Such a new structure is called a polarized $k$-symplectic structure.

In the language of $G$-structures, the pair $(\theta, \mathfrak{F})$ is equivalent to the existence on $M$ of a $Sp(k, n; \mathbb{R})$- structure [6], where $Sp(k, n; \mathbb{R})$ is the polarized $k$-symplectic group.

In this work, we study new geometrical and topological aspects of polarized $k$-symplectic manifolds. We give some properties of the De Rham cohomology group of the $k$-symplectic group and its Poincare group in order to highlight new topological properties of polarized $k$-symplectic geometry with respect to those of symplectic geometry and of the $k$-symplectic group with respect to the symplectic group, respectively.

It is well known that a symplectic manifold is orientable [2,7]; thus, we pay particular attention to the problem of the orientation of polarized $k$-symplectic manifolds in an analog way to symplectic manifolds by emphasizing other sides of $k$-symplectic manifolds.

## 2. Linear Polarized $k$-Symplectic Structures

*Definitions*

We denote by $\mathbb{K}$ the field of real numbers $\mathbb{R}$ or the field of complex numbers $\mathbb{C}$. Let $V = \mathbb{K}^k$ and $(v_1, \ldots, v_k)$ the canonical basis of $V$.

Let $E$ be a vector space of dimension $n(k+1)$ over $\mathbb{K}$, $F$ a vector subspace of codimension $n$ of $E$, and $\theta = \sum_{p=1}^k \theta^p \otimes v_p$ an exterior 2-form with values in $V$.

**Definition 1** ([3,5])**.** *The pair $(\theta, F)$ is said to be a polarized $k$-symplectic structure on $E$ if the following conditions are satisfied:*

1. *$\theta$ is not degenerate for every $x \in E, i_x\theta = 0 \Longrightarrow x = 0$.*
2. *$F$ is totally isotropic,*

$$\forall x, y \in F, \theta(x, y) = 0.$$

*We denote by $A(\theta)$ the subspace associated to the 2-vector form $\theta$ :*

$$A(\theta) = \{x \in E \mid i(x)\theta = 0\},$$

*then*

- *$\theta$ is non-degenerate if and only if, $A(\theta^1) \bigcap \cdots \bigcap A(\theta^k) = 0$;*
- *$F$ is totally isotropic if and only if, $\theta^p(x, y) = 0$ for every $x, y \in F$ and $p = 1, \ldots, k$.*

**Example 1.** *Let $E = \mathbb{R}^{n(k+1)}$, $(e_{pi}, e_i)_{1 \le p \le k, 1 \le i \le n}$ its canonical basis, and $(\omega^{pi}, \omega^i)_{1 \le p \le k, 1 \le i \le n}$ the dual basis of the canonical basis, and let $F$ be the vector subspace of $E$ generated by the vectors $(e_{pi})_{1 \le p \le k, 1 \le i \le n}$. For every $p = 1, \ldots, k$, we set*

$$\theta^p = \sum_{i=1}^n \omega^{pi} \wedge \omega^i$$

*and $\theta = \sum_{p=1}^k \theta^p \otimes v_p$.*

*The pair $(\theta, F)$ is a polarized $k$-symplectic structure on $E$.*

**Example 2.** *Consider $k$-symplectic vector spaces $(E^1, \sigma^1), \ldots, (E^k, \sigma^k)$ of the same dimension $2n$, and for each $p = 1, \ldots, k$, we consider a Lagrangian subspace $L^p$ of $(E^p, \sigma^p)$, i.e., a maximal totally isotropic subspace of this space . Each quotient vector space $E^p / L^p$ is of dimension $n$, and there exists an $n$-dimensional vector space $B$ and surjective linear maps $\pi^p : E^p \to B$ such that $\ker \pi^p = L^p$ for all $p = 1, \ldots, k$.*

*Consider the product symplectic space, $(E^1 \times \cdots \times E^k, \sigma)$, where $\sigma$ is the symplectic form*

$$\sigma((x_1, \ldots, x_k), (y_1, \ldots, y_k)) = \sigma^1(x_1, y_1) + \ldots + \sigma^k(x_k, y_k).$$

*Let E be the vector subspace of $E^1 \times \cdots \times E^k$ defined by*

$$E = \left\{ (x_1, \ldots, x_k) \in E^1 \times \cdots \times E^k \mid \pi^1(x_1) = \ldots = \pi^k(x_k) \right\}.$$

*The linear map $\pi : E \longrightarrow B$ such that $\pi(x_1, \ldots, x_k) = \ldots = \pi^k(x_k)$ is surjective and verifies $\ker \pi = L^1 \times \cdots \times L^k$. Let $i : E \longrightarrow E^1 \times \cdots \times E^k$ be the canonical injection, and let $\mathrm{pr}^p (p = 1, \ldots, k)$ be the canonical projection $E^1 \times \cdots \times E^k \longrightarrow E^p$. For all $p(p = 1, \ldots, k)$, the composition $\mathrm{pr}^p \circ i$ is the restriction of $\mathrm{pr}^p$ to E; it is a surjective linear map. It is clear that the space E is of dimension $n(k+1)$. For all $p = 1, \ldots k$, we set $\theta^p = (\mathrm{pr}^p \circ i)^* \sigma^p$, and $\theta = \sum_{p=1}^{k} \theta^p \otimes v_p$ . Then $(\theta, F)$ is a polarized k-symplectic structure on E.*

**Theorem 1** ([5]). *If $(\theta, F)$ is polarized k-symplectic structure on E with $\theta = \sum_{p=1}^{k} \theta^p \otimes v_p$, then there is a basis $(\omega^{pi}, \omega^i)_{1 \leq p \leq k, 1 \leq i \leq n}$ of $E^*$ such that*

$$\theta^p = \sum_{i=1}^{n} \omega^{pi} \wedge \omega^i, F = \ker \omega^1 \bigcap \ldots \bigcap \ker \omega^n.$$

*The basis $(e_{pi}, e_i)_{1 \leq p \leq k, 1 \leq i \leq n}$ of E with the dual basis $(\omega^{pi}, \omega^i)_{1 \leq p \leq k, 1 \leq i \leq n}$ is called a polarized k-symplectic basis on E.*

For $k \geq 2$ and $p = 1, \ldots k$, we take

$$F^p = \bigcap_{l \neq p} A(\theta^l).$$

Under the above assumptions and notations, we find the following:

1.  $F = F^1 \oplus \ldots \oplus F^k$;
2.  For every $p(p = 1, \ldots, k)$, the map $i_p : x \mapsto i(x)$ defines an isomorphism of vector spaces from $F^p$ to the annihilator $\mathrm{Ann}(F)$ of $F$. Recall that $\mathrm{Ann}(F)$ is formed by the elements $f$ of $E^*$ such that $f(x) = 0$, for all $x \in F$. This space is isomorphic to $(E/F)^*$. If $(e_{pi}, e_i)_{1 \leq p \leq k, 1 \leq i \leq n}$ is a k-symplectic basis of E and $(\omega^{pi}, \omega^i)_{1 \leq p \leq k, 1 \leq i \leq n}$ its dual basis, then $\mathrm{Ann}(F)$ is generated by $\omega^1, \ldots, \omega^n$, and $F^p$ is generated by the vectors $e_{p1}, \ldots, e_{pn}$.

The sub-spaces $F^1, \ldots, F^k$ are called characteristic sub-spaces of the polarized k-symplectic structure. For all $p = 1, \ldots, k$, we set

$$G^p = F^p \oplus B$$

and $B = E/F$. We have
$$\theta^p(x, y + f) = \theta^p(x, y),$$

for all $x \in F^p$, $y \in E$ and $f \in F$; then, $\theta^p(x, y)$ depends only on the class $\bar{y}$ of $y$ modulo $F$. This allows us to set
$$\bar{\theta}(x + \bar{y}, x' + \bar{y}') = \theta^p(x, y') - \theta^p(x', y),$$

where $\bar{\theta}^p$ defines a symplectic structure on $G^p = F^p \oplus B$.

Let E be a vector space of dimension $n(k+1)$ and $(\theta, F)$ a polarized k-symplectic structure on E with $\theta = \sum_{p=1}^{k} \theta^p \otimes v_p$.

**Definition 2** ([5]). *Let f be an endomorphism of E. We say that f preserves $(\theta, F)$ if it leaves invariant $\theta$ and the subspace F, i.e., if the following conditions are satisfied:*

1.  $f(F) \subseteq F$;
2.  $\forall x, y \in E, \theta(f(x), f(y)) = \theta(x, y), f^*\theta = \theta.y.$

Note that the non-degeneracy of $\theta$ implies that each endomorphism $f$ of E is an automorphism. The automorphisms of E which preserve the polarized k-symplectic

structure of $E$ form a group denoted $Sp(k, n; E)$ and called the *k-symplectic group* of $E$. Let $Sp(k, n; \mathbb{K})$ the group of matrices of *k*-symplectic automorphisms of $E$, expressed on a polarized *k*-symplectic basis, consist of the matrices of the type

$$
\begin{pmatrix}
Q & 0 & \cdots 0 & S_1 \\
0 & \ddots & & \vdots \\
\vdots & \ddots & Q & S_k \\
0 & \cdots & 0 & (Q^{-1})^T
\end{pmatrix}
\tag{2}
$$

where $Q, S_1, \ldots, S_k$ are real square matrices of order $n$ with coefficients in $\mathbb{K}$, $Q$ invertible and $QS_p^T = S_p Q^T$ for all $p(p = 1, \ldots, k)$. $Sp(k, n; E)$ is a Lie group.

All matrices of type (2) can be simply denoted by

$$
[(Q, S_1, \ldots, S_k)].
$$

The Lie algebra of this group, denoted by $\mathfrak{sp}(k, n; E)$ and which is identified with the tangent space of this group at the identity mapping of $E$, consists of the endomorphisms $u$ of $E$ such that

$$
u(F) \subseteq F, \; \theta(u(x), y) + \theta(x, u(y)) = 0
$$

for all $x, y \in E$.

Let $sp(k, n; \mathbb{K})$ be the Lie algebra of the Lie group $Sp(k, n; \mathbb{K})$; it is the Lie algebra of the matrices of the endomorphisms belonging to $\mathfrak{sp}(k, n; E)$ relative to the polarized *k*-symplectic basis. The elements of $sp(k, n; \mathbb{K})$ are all matrices of the type

$$
\begin{pmatrix}
A & 0 & \cdots 0 & S_1 \\
0 & \ddots & & \vdots \\
\vdots & \ddots & A & S_k \\
0 & \cdots & 0 & -A^T
\end{pmatrix}
$$

where $A, S_1, \ldots, S_k$ are square matrices of order $n$ with entries in $\mathbb{K}$ such that $S_p^T = S_p$ for all $p(p = 1, \ldots, k)$.

We observe the following:

1. The group $Sp(k, n; E)$ is of dimension $n^2 + \frac{kn(n+1)}{2}$;
2. The $k-$symplectic group and its Lie algebra leave invariant the characteristic subspaces of the polarized *k*-symplectic structure;
3. If $(\theta, F)$ is a polarized *k*-symplectic structure on an $n(k+1)$-dimensional vector space with $\theta = \sum_{p=1}^{k} \theta^p \otimes v_p$, then the forms $\theta^1, \ldots, \theta^k$ are of rank $2n$.

In the above assumptions and notations, we consider the space $G^p = F^p \oplus B$, $p = 1, \ldots, k$, endowed with the symplectic structure:

$$
\theta^p(x + y, x' + y') = \theta^p(x, y') - \theta^p(x', y),
$$

for all $x + y, x' + y' \in G^p$.

For every $f \in Sp(k, n; E)$, the map

$$
f_p : x + y \mapsto f(x) + f(y)
$$

defines an element of the polarized symplectic group $Sp(\theta^p, G^p, F^p)$ of the polarized symplectic vector space $(G^p, F^p, \theta^p)$.

### 3. Topological Properties of $Sp(k, n; E)$

We propose here to highlight new topological properties of the polarized $k$-symplectic geometry compared to those of the symplectic geometry through the $k$-symplectic and symplectic groups, respectively.

#### 3.1. Symplectic Group

Recall that a symplectic structure on a finite-dimensional vector space $E$ on $\mathbb{K}$, is defined by a non-degenerate exterior 2-form. The vector space $E$ is necessarily even-dimensional. Thus, a symplectic vector space is a pair $(E, \omega)$ in which $E$ is a vector space of dimension $2n$ and $\omega^n = \omega \wedge \cdots \wedge \omega$, ($n$-times), is a volume form on $E$.

There exists a basis $B = (e_1, \ldots, e_n, e'_1, \ldots, e'_n)$ in which the matrix of the 2-form $\omega$ is

$$J = \begin{pmatrix} 0 & I_n \\ -I_n & 0 \end{pmatrix},$$

or,

$$\omega = \sum_{i=1}^{n} \omega_i \bigwedge \omega'_i,$$

with $(\omega_1, \ldots, \omega_n, \omega'_1, \ldots, \omega'_n)$ the dual basis of $(e_1, \ldots, e_n, e'_1, \ldots, e'_n)$.

The basis $B$ is called the symplectic basis of $E$.

By virtue of the non-degeneracy of $\omega$, we deduce that, if an endomorphism $u$ of $E$ leaves invariant $\omega$, i.e., if $u^* \omega = \omega$, then $u$ is an automorphism of $E$.

The automorphisms of $E$ that leave $\omega$ invariant form a group denoted $Sp(E, \omega)$ and called the symplectic group of $E$.

Let $Sp(n, \mathbb{K})$ be the group of matrices of symplectic automorphisms of $E$ expressed on a symplectic basis. Then,

$$Sp(n, \mathbb{K}) = \left\{ P \in \mathrm{Gl}_{2n}(\mathbb{K}) \mid P^T J P = J \right\}.$$

In the case $\mathbb{K} = \mathbb{R}$, $Sp(E, \omega)$ is a Lie group of dimension $n(2n + 1)$, and its Lie algebra is denoted by $\mathfrak{sp}(E, \omega)$—it consists of endomorphisms $f$ of $E$ such that

$$\omega(f(x), y) + \omega(x, f(y)) = 0,$$

for all $x, y \in E$. Let $\mathfrak{sp}(n, \mathbb{R})$ be the Lie algebra of $Sp(n, \mathbb{R})$; then,

$$\mathfrak{sp}(n, \mathbb{R}) = \left\{ A \in \mathfrak{gl}_{2n}(\mathbb{R}) \mid A^T J = -J A \right\}.$$

Let us denote by $O_{2n}(\mathbb{R})$ the orthogonal group, $U_n(\mathbb{C})$ the group of unitary matrices of order $n$, and $SU_n(\mathbb{C})$ the unitary special group:

$$
\begin{aligned}
O_{2n}(\mathbb{R}) &= \left\{ A \in Gl_{2n}(\mathbb{R}) \mid A^T = A^{-1} \right\}, \\
U_n(\mathbb{C}) &= \left\{ A \in Gl_n(\mathbb{C}) \mid \overline{A}^T = A^{-1} \right\}, \\
SU_n(\mathbb{C}) &= \left\{ A \in U_n(\mathbb{C}) \mid \det A = 1 \right\}.
\end{aligned}
$$

By well-known results on algebraic groups, the $Sp(n, \mathbb{R})$ group is homeomorphic to $(O_{2n}(\mathbb{R}) \bigcap Sp(n, \mathbb{R})) \times \mathbb{R}^m$, and so

$$Sp(n, \mathbb{R}) \simeq (O_{2n}(\mathbb{R}) \bigcap Sp(n, \mathbb{R})) \times \mathbb{R}^m,$$

for a certain natural number $m$. In addition, it is well known that

$$
\begin{aligned}
O_{2n}(\mathbb{R}) \bigcap Sp(n, \mathbb{R}) &= \left\{ M = \begin{pmatrix} A & -B \\ B & A \end{pmatrix} \mid A, B \in \mathfrak{gl}_n(\mathbb{R}) \ \text{et}\ M \in O_{2n}(\mathbb{R}) \right\} \\
&\simeq \qquad\qquad U_n(\mathbb{C})
\end{aligned}
\tag{3}
$$

and therefore,

$$Sp(n, \mathbb{R}) \simeq U_n(\mathbb{C}) \times \mathbb{R}^m$$

Therefore, $Sp(n, \mathbb{R})$ and $U_n(\mathbb{C})$ have the same topological invariants, so in particular, $Sp(n, \mathbb{R})$ and $U_n(\mathbb{C})$ have the same homotopy groups and have the same cohomology groups. In particular,

- $Sp(n, \mathbb{R})$ is connected;
- The Poincare group: $\pi_1(Sp(n, \mathbb{R})) = \mathbb{Z}$;
- De Rham cohomology groups: $H^1(Sp(n, \mathbb{R})) = \mathbb{R}$ ; $H^2(Sp(n, \mathbb{R})) = \{0\}$ ; $H^{n^2}$ $(Sp(n, \mathbb{R})) = \mathbb{R}$ and $H^p(Sp(n, \mathbb{R})) = \{0\}$ for $p > n^2$.

**Remark 1.** *If we look at $U_n(\mathbb{C})$, via (2), as a subgroup of $Sp(n, \mathbb{R})$, then the homogeneous space $Sp(n, \mathbb{R})/U_n(\mathbb{C})$ is contractile.*

*3.2. Topological Properties of $Sp(k, n; E)$*

We denote by $S_n(\mathbb{K})$ the vector space of the symmetric square matrices of order $n$. The $k$-symplectic group is

$$\begin{pmatrix} Q & 0 & \cdots 0 & S_1 \\ 0 & \ddots & & \vdots \\ \vdots & \ddots & Q & S_k \\ 0 & \cdots & 0 & (Q^{-1})^T \end{pmatrix} = [(Q, S_1, \ldots, S_k)]$$

with $Q \in GL_n(\mathbb{K})$, $S_1, \ldots, S_k \in \mathfrak{gl}_n(\mathbb{K})$ and $S_p Q^T \in S_n(\mathbb{K})$.

From this, it is immediately clear that we have

**Proposition 1.** *The group $Sp(k, n; \mathbb{K})$ is diffeomorphic to $GL_n(\mathbb{K}) \times (S_n(\mathbb{K}))^k$.*

**Proof.** The map

$$[(Q, S_1, \ldots, S_k)] \longmapsto (Q, S_1 Q^T, \ldots, S_k Q^T)$$

of $Sp(k, n; \mathbb{K})$ into $GL_n(\mathbb{K}) \times (S_n(\mathbb{K}))^k$ is a diffeomorphism of $Sp(k, n; \mathbb{K})$ to $GL_n(\mathbb{K}) \times (S_n(\mathbb{K}))^k$. □

In particular, $Sp(k, n; \mathbb{K})$ is diffeomorphic to $GL_n(\mathbb{K}) \times (S_n(\mathbb{K}))^k$, and therefore, $Sp(k, n; \mathbb{K})$ is of the same type of homotopy as $GL_n(\mathbb{K})$.

We have the following cases:

1. Case $\mathbb{K} = \mathbb{C}$: The polar decomposition and exponential application show that $GL_n(\mathbb{C})$ is homeomorphic to $U_n(\mathbb{C}) \times H_n(\mathbb{C})$, where

$$H_n(\mathbb{C}) = \left\{ A \in \mathfrak{gl}_n(\mathbb{C}) \mid \overline{A}^T = A \right\}$$

the real vector space of Hermitian matrices, so $Sp(k, n; \mathbb{C})$ is of the same homotopy type as $U_n(\mathbb{C})$, so it has the same topological properties.

2. Case $\mathbb{K} = \mathbb{R}$.

   (a) $Sp(k, n; \mathbb{R})$ has two connected components $Sp^+(k, n; \mathbb{R})$ and $Sp^-(k, n; \mathbb{R})$, where

   $$Sp^+(k, n; \mathbb{R}) = \{[(Q, S_1, \ldots, S_k)] \in Sp(k, n; \mathbb{R}) \mid \det(Q) > 0\}$$

   and

   $$Sp^-(k, n; \mathbb{R}) = \{[(Q, S_1, \ldots, S_k)] \in Sp(k, n; \mathbb{R}) \mid \det(Q) < 0\}.$$

   (b) By a similar reasoning to the complex case,

   $$GL_n(\mathbb{R}) \simeq O_n(\mathbb{R}) \times S_n(\mathbb{R}),$$

where $S_n(\mathbb{R})$ is the vector space of symmetric real matrices. Therefore, the Poincare group

$$\pi_1(Sp(k,n;\mathbb{R})) = \begin{cases} \{0\} \text{ if } n = 1 \\ \mathbb{Z} \text{ if } n = 2 \\ \mathbb{Z}_2 \text{ if } n \geq 3 \end{cases}$$

with $\mathbb{Z}_2$ is the two-elements group. Therefore, $Sp^+(k,n;\mathbb{R})$ is simply connected if and only $n = 1$ (even contractile).

(c) Here are some de Rham cohomology groups:

$$H^0(Sp(k,n;\mathbb{R})) = \mathbb{R}^2 (\forall n \in \mathbb{N}^*),$$

and for $n = 1$, $n = 2$ and $n \geq 3$, we have

i. $H^p(Sp(k,1;\mathbb{R})) = \{0\}, \forall p \geq 1,$

ii. $H^1(Sp(k,2;\mathbb{R})) = \mathbb{R}^2$ and $H^p(Sp(k,2;\mathbb{R})) = \{0\}, \forall p \geq 2,$

iii. $H^1(Sp(k,n;\mathbb{R})) = H^2(Sp(k,n;\mathbb{R})) = \{0\}$, $H^{((n(n-1))/2)}(Sp(k,n;\mathbb{R})) = \mathbb{R}^2$ and $H^p(Sp(k,n;\mathbb{R})) = \{0\}$, for every $n \geq 3$ and $p > ((n(n-1))/2)$.

## 4. $k$-Symplectic Action of $Sp(k,n;\mathbb{K})$

We identify $\mathbb{K}^m$ with $\mathcal{M}_{m,1}(\mathbb{K})$ the space of the column matrices. Consider the natural action of the group $Sp(k,n;\mathbb{K})$ on $\mathbb{K}^{nk} \times (\mathbb{K}^n - (0_{\mathbb{K}^n}))$. $S_n(\mathbb{K})$ consists of vector space of symmetric matrices with entries in $\mathbb{K}$.

**Lemma 1.** *Let $X \in \mathbb{K}^n - (0_{\mathbb{K}^n})$, then, the map $A \mapsto AX$, from $S_n(\mathbb{K})$ into $\mathbb{K}^n$, is surjective.*

**Proof.** First, let the column vector $X = E_1 = (1, 0, \cdots, 0)^T$. For each $A \in S_n(\mathbb{K})$, the column vector $AX$ is the first column of matrix $A$. As any vector of $\mathbb{K}^n$ can be the first column of a symmetric matrix, then this map is surjective.

For any vector $X \in \mathbb{K}^n - (0_{\mathbb{K}^n})$, there is $P \in GL_n(\mathbb{K})$ such that $PX = E_1$. Let $Y \in \mathbb{K}^n$; then, there exists a matrix $B \in S_n(\mathbb{K})$ such that $BE_1 = (P^{-1})^T Y$. Then, we obtain $P^T BPX = Y$ with $P^T BP \in S_n(\mathbb{K})$, and this proves the result. $\square$

**Proposition 2.** *$Sp(k,n;\mathbb{K})$ acts transitively on $\mathbb{K}^{nk} \times (\mathbb{K}^n - (0_{\mathbb{K}^n}))$*

**Proof.** Let $\xi = (X_1, \ldots, X_k, X)^T$ and $\xi' = (X_1', \ldots, X_k', X')^T$, with $X_p, X_p' \in \mathbb{K}^n$ for every $p = 1, \ldots, k$ and $X, X' \in \mathbb{K}^n - (0_{\mathbb{K}^n})$. We must find.

$M = [(Q, S_1, \ldots, S_k)] \in Sp(k,n;\mathbb{K})$ such that $M\xi = \xi'$. Then we must have, $(Q^{-1})^T X = X'$ and $QX_p + S_p X = X_p'$ for every $p = 1, \ldots, k$.

As $X, X' \in \mathbb{K}^n - (0_{\mathbb{K}^n})$, then, there exists $P \in GL_n(\mathbb{K})$ such that $PX = X'$. So, we take $Q = (P^{-1})^T$. The above lemma justifies the existence of matrices $A_1, \ldots, A_k \in S_n(\mathbb{K})$ such that $A_p(Q^{-1})^T X = X_p' - QX_p$ for each $p = 1, \ldots, k$, also, we take $S_p = A_p(Q^{-1})^T$. $\square$

**Corollary 1.** *Let $(\theta, F)$ be a polarized $k$-symplectic structure on $E$. The $k$-symplectic group $S_p(k,n;E)$ acts transitively on $E$–$F$.*

**Corollary 2.** *In the case where $\mathbb{K} = \mathbb{R}$, we have :*

1. *$Sp^+(k,n;\mathbb{R})$ acts transitively on $\mathbb{K}^{nk} \times (\mathbb{K}^n - (0_{\mathbb{K}^n}))$, if $n \geq 2$.*

2. *The action of $Sp^+(1,n;\mathbb{R})$ on $\mathbb{R}^k \times \mathbb{R}^*$, admits two orbits $\mathbb{R}^k \times \mathbb{R}^{*+}$ and $\mathbb{R}^k \times \mathbb{R}^{*-}$, if $n = 1$.*

**Proof.** We denote by $GL^+(\mathbb{R})$ the set of invertible real matrices of positive determinants and we resume the previous proof of the fact that

1. If $n \geq 2$, $GL^+(\mathbb{R})$ acts transitively on $\mathbb{R}^n - \{0\}$,

2. If $n = 1$, the action of $GL_1^+(\mathbb{R}) = \mathbb{R}^{*+}$ on $\mathbb{R} - \{0\}$ induces two orbits $\mathbb{R}^{*+}$ and $\mathbb{R}^{*-}$.

$\square$

## 5. Non-Orientable Polarized *k*-Symplectic Manifolds

### 5.1. Polarized k-Symplectic Manifolds

Let $M$ be a differentiable manifold of dimension $n(k+1)$ equipped with a foliation $\mathfrak{F}$ of codimension $n$ and let $\theta \in \mathscr{A}^2(M) \otimes \mathbb{R}^k$ be a differential 2-form over $M$ with values in $\mathbb{R}^k$. We denote by $E$ the sub-bundle of $TM$ defined by the vectors tangent to the leaves of $\mathfrak{F}$.

**Definition 3.** *We say that* $(M, \theta, \mathfrak{F})$ *is a polarized k-symplectic manifold if the following hold:*

(i)   *$\theta$ is closed ($d\theta = 0$);*
(ii)  *$\theta$ is non degenerate;*
(iii) *$\theta(X, Y) = 0$ for every vector field $X, Y$ tangent to $\mathfrak{F}$: $X, Y \in T(\mathfrak{F})$.*

Recall here the theorem of the local model of the Darboux type with respect to polarized *k*-symplectic structures [5].

**Theorem 2.** *If $(\theta, E)$ is a polarized k-symplectic structure on $M$ with $\theta = \sum_{p=1}^{k} \theta^p \otimes v_p$, then around each point $x_0$ of $M$, there is an open neighborhood $U$ of $M$ containing $x_0$ local coordinates $(x^{pi}, y^i)_{1 \leq p \leq k, 1 \leq i \leq n}$ called adapted, such that the vectorial 2-form $\theta$ is represented in $U$ by:*

$$\theta_{|U} = \left( \sum_{i=1}^{n} dx^{pi} \wedge dy^i \right) \otimes v_p,$$

*and the sub-bundle $E_{|U}$ is defined by the equations $dy^1 = \ldots = dy^n = 0$.*

The coordinate changes in this atlas are given by

$$\overline{x}^{pi} = \sum_{j=1}^{n} \frac{\partial y^j}{\partial \overline{y}^i} x^{pj} + \varphi^{pi}\left(y^1; \cdots, y^n\right) \ , \ \ \overline{y}^i = \overline{y}^i\left(y^1; \cdots, y^n\right).$$

Equivalently, there is an atlas $\mathfrak{A}$ of $M$, called the polarized *k*-symplectic atlas, whose coordinate changes belong to the pseudogroup of local diffeomorphisms of $\mathbb{R}^{n(k+1)}$, which is a canonical invariant of a polarized *k*-symplectic structure.

**Definition 4.** *A k-symplectomorphism of a k-symplectic manifold $(M, \theta, \mathfrak{F})$ on a k-symplectic manifold $(M', \theta', \mathfrak{F}')$ is a diffeomorphism $f$ of $M$ on $M'$ that exchanges the k-symplectic structures $(\theta, \mathfrak{F})$ and $(\theta', \mathfrak{F}')$, i.e., $f^*\theta' = \theta$ and $f(\mathfrak{F}) = \mathfrak{F}'$.*

### 5.2. Orientation

It is well known that a symplectic manifold $(N, \omega)$ of dimension $2n$ is orientable, because $\omega^n$ is a volume form on $N$.

Let $(M, \theta, \mathfrak{F})$ be a *k*-symplectic manifold.

**Proposition 3.** *If $k$ is odd, then $M$ is orientable.*

**Proof.** By virtue of the Darboux theorem of the local model for polarized *k*-symplectic structures [5], we can find an atlas of $M$ whose Jacobian matrices $\underline{U}$ of map changes, belong to $Sp(k, n; \mathbb{R})$, i.e., matrices of the type

$$U = \begin{pmatrix} Q & 0 & \cdots 0 & S_1 \\ 0 & \ddots & & \vdots \\ \vdots & \ddots & Q & S_k \\ 0 & \cdots & 0 & (Q^{-1})^T \end{pmatrix} = [(Q, S_1, \ldots, S_k)],$$

with $Q \in GL_n(\mathbb{R})$, $S_1, \ldots, S_k \in \mathfrak{gl}_n(\mathbb{R})$ and $S_p Q^T = Q S_p^T$, so,

$$\det U = (\det Q)^{k-1} > 0.$$

$\square$

Suppose that $k$ is even. In this casel the determinants of the two connected components of $Sp(k, n; \mathbb{R})$ have opposite signs.

In the following, we propose to give examples of non-orientable $k$-symplectic manifolds by studying the quotient covering obtained by discontinuous actions of a group of diffeomorphisms on a $k$-symplectic manifold.

Let $X$ be a differentiable manifold and $G$ a subgroup of $\mathrm{Diff}(X)$the group of diffeomorphisms of $X$.

Recall that $G$ acts properly and discontinuously without a fixed point on $X$ if the following two properties are satisfied:

1. For any $x \in X$, there is an open $U$ of $X$ containing $x$ such that for any $g \in G - \{id_X\}$, $g(U) \cap U = \varnothing$.
2. For any $x, y \in X$, such that $y$ is not in the orbit of $x$, there exists a neighborhood $U$ of $x$ and a neighborhood $V$ of $y$ such that $g(U) \cap V = \varnothing$ for any $g \in G$.

**Theorem 3.** *Under the assumptions of the previous definition, there exists on the quotient space $X/G$ of the orbits a single structure of differentiable manifold such that the canonical projection $p$ is a differentiable covering. In this case, the fibers are all isomorphic to $G$.*

Under the conditions of the previous theorem, we find the following:

1. If $X$ is simply connected, then the Poincare group $\pi_1(X/G)$ is isomorphic to $G$;
2. Let $G$ be a finite subgroup of $\mathrm{Diff}(X)$; then, $G$ acts properly discontinuously without a fixed point on $X$ if and only if, for any $g \in G - \{id_X\}$, $g$ is without a fixed point.

**Theorem 4.** *If $G$ is a subgroup of the group of k-symplectomorphisms of the polarized k-symplectic manifold $(M, \theta, \mathfrak{F})$, acting properly and discontinuously without a fixed point on $M$, then $M/G$ admits a unique structure of polarized k-symplectic manifold such that the covering $p$ is a local k-symplectomorphism.*

**Proof.** The structure of the differentiable manifold on $M/G$ defined above admits a polarized $k$-symplectic structure which is locally defined as an image by the covering $p$, and this definition is intrinsic since the elements of $G$ preserve the form $\theta$ and the foliation $\mathfrak{F}$. $\square$

**Theorem 5.** *Let $X$ be a differentiable manifold and $G$ a subgroup of the group $\mathrm{Diff}(X)$ which acts properly and discontinuously without a fixed point on $X$.*

1. *If $X$ is orientable and the elements of $G$ preserve an orientation, then the manifold $X/G$ is orientable;*
2. *If $X$ is connected and orientable, then $X/G$ is orientable if and only if the elements of $G$ preserve an orientation.*

**Proof.** 1. We consider the canonical projection $p : X \to X/G$ and let $(U_\alpha, \phi_\alpha)_{\alpha \in I}$ be an atlas of $X$, which defines an orientation preserved by the elements of $G$, where

$$p_{/U_\alpha} := p_\alpha : U_\alpha \longrightarrow p(U_\alpha)$$

is a diffeomorphism, for each $\alpha \in I$; then, we verify immediately that the atlas $\left( p(U_\alpha), \phi_\alpha \circ p_\alpha^{-1} \right)_{\alpha \in I}$ defines an orientation on $X/G$.

2. Note that, from the connectedness of $X$, we deduce that if an element of $\mathrm{Diff}(X)$ conserves an orientation of $X$, then it preserves any orientation of $X$. It is a question of showing that if $X/G$ is orientable, then the elements of $G$ preserve an orientation.

If $\omega$ is a volume form of $X/G$, then $p^*\omega$ is a volume form of $X$. We suppose there is an element $g \in G$ that does not preserve an orientation; then, there exists $f \in C^\infty(X)$ which takes strictly negative values such that $g^*[p^*\omega] = f(p^*\omega)$, but $p \circ g = p$. Therefore, $g^*[p^*\omega] = (p \circ g)^*\omega = p^*\omega$, which is absurd. $\quad\square$

*5.3. Non-Orientable Polarized k-Symplectic Manifolds*

1.  Let $k, n \in \mathbb{N}$. For $n$ odd and $k$ even, we consider the space $M = \mathbb{R}^{n(k+1)} - \{0\}$ equipped with its standard polarized $k$-symplectic structure. The set $\{id, -id\}$ is a subgroup of $k$-symplectomorphisms. As $M$ is connected and $-id$ does not preserve the orientation, then $\frac{M}{\{id,-id\}}$ is a non-orientable polarized $k$-symplectic manifold which is diffeomorphic to $\mathbb{R} \times \mathbb{RP}(m)$ where $m = nk + n - 1$;

2.  We assume that $k$ is even and $n \geq 2$. We consider an invertible diagonal matrix

$$Q = \mathrm{diag}(\lambda_1, \ldots, \lambda_n) \in Gl_n(\mathbb{R}),$$

where, $\lambda_1, \ldots, \lambda_{n-1} > 1$ and $\lambda_n < -1$. We take

$$L = [(Q, 0_n, \ldots, 0_N)] \in Sp(k, n; \mathbb{R}).$$

Let $M = \left(\mathbb{R}^{nk} - \{0_{\mathbb{R}^{nk}}\}\right) \times (\mathbb{R}^n - \{0_{\mathbb{R}^n}\})$ the standard connected polarized $k$-symplectic manifold. The subgroup $\langle L \rangle$ generated by $L$ is a subgroup of the group of $k$-symplectomorphisms that acts properly and discontinuously without a fixed point on $M$. $L$ does not preserve the orientation, so the quotient $\frac{M}{\langle L \rangle}$ is a non-orientable polarized $k$-symplectic manifold.

**Proposition 4.** *Let $S^m$ be the sphere of dimension m and k an odd integer. The sphere $S^{m(k+1)}$ does not support any structure of a polarized k-symplectic manifold such that $-id$ is a k-symplectomorphism.*

**Proof.** If $-id$ is a $k$-symplectomorphism, then the projective space $\frac{S^{n(k+1)}}{\{id,-id\}}$ of even dimension is a polarized $k$-symplectic manifold where $k$ is odd and therefore orientable, which is absurd, because $-id$ preserves the orientation of $S^m$ only if $m$ is odd. $\quad\square$

## 6. Conclusions

In this paper, We give some properties of the De Rham cohomology group of the $k$-symplectic group and its Poincare group in order to highlight new topological properties of polarized $k$-symplectic geometry with respect to those of symplectic geometry and of the $k$-symplectic group with respect to the symplectic group, respectively. We have shown that, unlike symplectic manifolds, the polarized $k$-symplectic manifolds are not all orientable.

**Author Contributions:** Conceptualization, E.S.; Methodology, A.A.; Validation, A.A.; Writing—original draft, E.S.; Writing—review & editing, E.S. and F.E.M. All authors have read and agreed to the published version of the manuscript.

**Funding:** This research received no external funding.

**Institutional Review Board Statement:** Not applicable.

**Informed Consent Statement:** Not applicable.

**Data Availability Statement:** Not applicable.

**Conflicts of Interest:** The authors declare that there is no conflict of interests.

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
