# Peer review of "Topological and Geometrical Properties of k-Symplectic Structures"

_axioms, doi:10.3390/axioms11060273_

Round 1

Reviewer 1 Report

The paper is of some interest and I suggest publication but there are many typos and must be carefully checked again:

row +2 of abstract: Rham instead of RHAM and delete one "of"

page 3, row +3 and +5, correct \alpha

page 6, 3.1 Symplectic instead of symplectic and row -3 remove points .J.P

page 7, row -6 delete one point

page 8 row +2 We denote instead of We note

page 11, row +14 delete ";"

page 13, +19 check "differentomorphic"

Author Response

Veuillez consulter le fichier ci-joint s'il vous plaît

Reviewer 2 Report

Comment. The symplectic structure is a fundamental one, with important applications in the 
modeling of physical phenomena. Its generalizations are susceptible to be able to refine the 
classical theories.
Conclusion. The paper may be published after minor revisions.
Motivation.
1. The need of generalization from symplectic manifold to k-symplectic manifolds is claimed in 
the introduction. The authors must explain also, briefly, why polarization is important in 
applications, and must provide appropriate references.
2. The authors must clearly point out what is original and what is knew, firstly in the 
introduction and after that in the next sections. The known notions and results must be quoted
very precisely. For example, from the paper one deduce that “polarized k-symplectic structure” 
is new (page 2 line 7). However, the notion appears in some previous papers of the authors.
3. The title, the abstract and the introduction refers to “topological and geometrical properties” of 
k-symplectic manifolds. The topological, the algebraic topological and the differential 
topological properties are clear. Which are the geometrical properties the authors consider the 
most important?
4. There are many typos. In the attached pdf file I highlighted some of them. Especially, a 
comma between the subject and its verb must be avoided

Reviewer 3 Report

       In the present paper the authors studied linear polarized k-symplectic structures (section 2), tolopogical properties of polarized k-symplectic geometry (section 3) and k-symplectic action of k-symplectic group of E (section 4). The last section considers non-orientable polarized k-symplectic manifolds.

The aim of the paper is to give certain properties of de Rham cohomology group of the k-symplectic group and its Poincare group.

       I suggest:

11. the correction of several typos (are mentioned in the pdf file);

 2.  the improvement of the introduction and presentation. 

      I invite the authors to describe briefly the potential applications of the proposed methods. This would be of interest for the general reader. Please, include also related references.
